# Impact of combined medication payment management policies on population health performance

Dingqiang Duan, Yun Yang 📛 *

School of Economics, Wuhan Textile University, Wuhan, Hubei, China

* 992937871@qq.com

## Abstract

This study investigated the mechanism and impact of a policy combination involving centralized drug procurement and national drug price negotiations on health insurance payment management and overall health performance of the population. Relying on a two-dimensional analytical framework of health outcomes and medical expenditures, the entropy value method was applied to construct indicators of residents' health portfolios. The year 2019, marked by the large-scale implementation of centralized procurement and national medicine catalog negotiations, was identified as the policy breakpoint for constructing a breakpoint regression model. Based on CFPS data, the model was implemented to evaluate changes in residents' health outcomes and medical expenditure efficiency. Furthermore, the mechanisms underlying policy effects were examined from the perspectives of drug expenditure, pharmaceutical innovation (e.g., R&D inputs and patent output), and drug trade including imports and exports. The results indicated that this policy combination significantly improved population health outcomes and enhanced the efficiency of healthcare spending. Mechanism analysis further confirmed its short-term effects on stimulating innovation, increasing drug accessibility, and promoting expenditure efficiency. In addition, empirical evidence supported the hypothesized synergy between import substitution and export upgrading. Therefore, it is recommended to establish a value-oriented drug classification and payment management mechanism while adapting regional policies to provide a scientific basis for optimizing pharmaceutical policy design and balancing health accessibility with the advancement of innovation in the pharmaceutical industry.

## 1. Introduction

People's health serves as a fundamental benchmark for evaluating the effectiveness of Chinese-style modernization. The 2025 Chinese Government Work Report emphasizes the implementation of a health-first development strategy to enhance

**Data availability statement:** The data that support the findings of this study are derived from the China Family Panel Studies (CFPS). The CFPS data are collected and distributed by the Institute of Social Science Survey (ISSS) at Peking University. The original data used in this study can be accessed upon application through the official CFPS website (https://www.isss.pku.edu.cn/cfps/).

**Funding:** Humanities and Social Sciences Research Project of the Ministry of Education: "Mechanism and Policy Research on the Impact of Centralized Volume-Based Drug Procurement on China's Pharmaceutical Innovation Ecosystem" (22YJAZH014).

**Competing interests:** The authors have declared that no competing interests exist.

population well-being. Since 2018, the health insurance drug payment policy portfolio has centered on volume-based procurement, and the negotiation mechanism of the health insurance catalog has undergone systematic reform through the strategy of "phasing out low-value products and introducing high-value alternatives". This approach aims to rationalize drug prices, optimize the allocation of healthcare resources, and steer the pharmaceutical industry towards innovation and development through a dual mechanism of incentives and constraints [1]. According to statistics, the nationally organized volume-based procurement of drugs has cumulatively saved approximately 440 billion yuan in health insurance funds, of which over 360 billion yuan has been redirected to cover negotiated drugs. This reallocation, characterized by simultaneous cost reduction and quality improvement, has significantly enhanced the efficiency of health insurance fund utilization. The remaining savings primarily supported public hospital salary reform. This comprehensive policy framework not only utilizes price discovery to eliminate inflated generic drug prices but also leverages strategic fund allocation to facilitate the inclusion of innovative drugs in the catalog. Additionally, salary system reforms have influenced the financial link between prescription behavior and rebate-driven incentives. Through this three-pronged strategy, reductions in pharmaceutical expenditures, elevation of medical service value, and upgrading of industrial innovation capacity have been jointly achieved, realizing Pareto improvement and advancing the development of high-quality productivity in the pharmaceutical sector.

With the ongoing implementation of the Healthy China Strategy, public expectations for accessibility, affordability, and quality of medical services continue to rise. However, despite the slow growth in national medical expenditure in recent years, the financial burden on individuals has increased, and public concerns over the quality of rapidly substituted generic medicines have intensified. Therefore, it is of both theoretical and practical significance to conduct a scientific evaluation of whether the current medical insurance payment management policy combination enhances the efficiency of medical expenditures and improves health outcomes while also addressing public concerns and expectations.

In evaluating the performance of medication payment management policies, the existing literature has predominantly adopted a single-policy perspective, focusing on unidirectional impacts on the government [2], hospitals [3], and pharmaceutical companies [4]. However, such studies often lack a comprehensive assessment of policy combinations and fail to decompose and examine policy outcomes in terms of both performance and effectiveness. Consequently, these limitations hinder the capacity to fully address the objectives of the "Healthy China" strategy, aimed at enhancing public well-being and health. In response, this study developed a theoretical analytical framework of "policy tool-market response-health performance" and employed the entropy weight method to construct composite indicators reflecting both the "performance" and "effectiveness" of residents' health outcomes. Furthermore, regression analysis was applied to evaluate the outcomes of the integrated policy approach under the "Healthy China" strategy. Utilizing a breakpoint regression model and CFPS microdata, this study assessed the influence of these policies on

residents' health outcomes and economic burden, while further analyzing the underlying mechanisms through factors such as residents' pharmaceutical expenditures and pharmaceutical innovation (including R&D investment, patent output, and import-export activity of medicines).

The key innovations of this study are as follows. First, a two-dimensional evaluation system of "health benefits and healthcare expenditure efficiency" was analyzed to examine how policy combinations could improve health outcomes by reshaping the competitive structure of the pharmaceutical market and encouraging innovation. Second, it moved beyond the limitations of traditional closed-system evaluations, revealing the mechanisms of domestic innovation incentives (supply side) and global value chain embeddedness (distribution side) in shaping health outcomes. These findings provide new theoretical foundations for designing pharmaceutical payment policies in an open and competitive market environment.

## 2. Theoretical analysis and research hypotheses

### 2.1. Impact of medicare medication payment management policies on residents' health performance

Based on the concept of value-based healthcare, this study established a dual-dimensional framework for evaluating policy performance. The first dimension concerned residents' physical health benefits, encompassing physiological functions, disease conditions, and quality of life, to represent the core objectives of health policy. The second dimension focused on the efficiency of medical expenditure, reflecting the comprehensive nature of healthcare services, including accessibility, drug quality, and rational drug use.

**2.1.1. Drug price reduction affects patients' medication behavior.** Drug prices are a critical factor influencing patient medication decisions. Payment management policies that reduce prices contribute to improved medication adherence. Specifically, the inclusion of drugs in centralized volume-based procurement and national negotiation lists [5] significantly lowers prices, thereby easing the economic burden on patients, conserving medical insurance funds [6], and improving health outcomes [7]. Simultaneously, reduced drug prices can enhance patient adherence to treatment, minimizing the likelihood of medication reduction or discontinuation due to financial constraints [8], thereby preventing deterioration in health status. Nevertheless, lower costs may also trigger an excessive release of latent "health needs", potentially resulting in overutilization of low-value medical services [9] and diminished engagement in preventive health behaviors [10].

**2.1.2. Differences in accessibility of different categories of drugs due to centralized purchasing and health insurance catalogs.** The combination of centralized procurement and health insurance catalog negotiations improves drug accessibility by leveraging scale effects. The policy facilitates "exchange volume for price" arrangements [11], resolving issues such as the decoupling of price and volume, weak bargaining power in decentralized procurement, and lack of policy cohesion. Although such mechanisms enhance public welfare through price transfers [12], excessive price suppression may diminish pharmaceutical companies' incentives for R&D investment [13], potentially compromising the long-term quality and availability of innovative drugs [14]. Moreover, inadequate provision of innovative therapies in primary care coupled with hospitals' preference for low-priced options may result in insufficient access to high-quality drugs, thereby limiting the actual improvement in drug affordability and negatively affecting patient outcomes.

**2.1.3. Access gap under policy effectiveness.** This policy has yielded preliminary outcomes in curbing medical expenses by reducing drug distribution costs [15]. The composition of hospitalization expenditures has been optimized, with a marked decline in the proportion allocated to medications, thereby providing institutional support for alleviating patients' financial burden related to pharmaceuticals [16]. However, a discrepancy remains between the observed reduction in out-of-pocket expenses and the actual perception of access among residents. On the one hand, adjustments to payment standards for non-negotiated drugs have limited access. On the other hand, price reductions have released previously suppressed demand [17]. This mismatch between macro-level cost control and micro-level access perception has led to divergence between the policy's intended outcomes and residents' experiences.

Hypothesis 1: The combination of medication payment management policies improves population health performance.

## 2.2. Pharmaceutical innovation and health performance

China's health insurance drug payment management policy has a dual impact on the pharmaceutical innovation eco-system by reshaping market incentive structures. From the perspective of the domestic market, in response to the bio-pharmaceutical industry's long-standing emphasis on marketing over R&D [18], volume-based procurement policy has compelled enterprises to transition towards innovation-driven models by constraining the marketing scope of generic drugs and reducing the market share of originator drugs with expired patents [19]. However, the policy's incentive effect on innovation remains limited. Accelerating access to innovative drugs and providing payment protection through health insurance negotiations have been designed to establish clearer innovation return expectations, thereby promoting a grad-ual transition of the industry toward innovation-driven development.

In the international value chain dimension, policies have facilitated domestic substitution by lowering the prices of imported originator drugs [20]. Nonetheless, markets in technology-intensive fields and rare disease treatments remain heavily reliant on imports. While such imports help address accessibility issues for patented drugs, multinational pharma-ceutical corporations continue to reap excessive profits through patent monopolies [21], intensifying the medication bur-den in developing countries. In response, the Chinese government has implemented price controls and expanded national drug negotiations to reduce the economic burden on patients and improve therapeutic effectiveness [5]. This has led to the formation of a synergistic policy path of "import substitution-export upgrading". This policy-induced dual mechanism linking local innovation incentives with global value chain restructuring reflects the internal logic of reforms in the drug payment management system. It enhances health outcomes by optimizing the allocation of innovation-related resources.

Hypothesis 2: The health insurance medication payment policy mix enhances population health performance by reducing residents' drug expenditures, incentivizing pharmaceutical firms' R&D innovation, and leveraging synergies from import substitution and export upgrading within global value chains.

## 3. Materials and methods

### 3.1. Resident health performance indicators and measurement

**3.1.1. Indicator design.** To objectively assess residents' status, this study employed the entropy weight method to construct a composite index. This method determines weights objectively based on information entropy theory, effectively avoiding subjective arbitrariness and making it particularly suitable for a multi-indicator comprehensive evaluation.

The established indicator system encompassed two dimensions: "Performance" (health outcomes) and "Efficiency" (financial burden). The "Performance" dimension included self-rated health, incidence of sudden illness, and prevalence of chronic disease. The "Efficiency" dimension consisted of annual total medical expenditure, out-of-pocket spending, and the incidence of catastrophic health expenditures. To address scale differences and mitigate potential skewness, loga-rithmic transformation was applied to the two continuous variables: annual total medical expenditure and out-of-pocket spending. This comprehensive indicator system integrates multiple aspects, such as health perception, disease risk, and economic pressure, thereby providing a holistic measure of policy effectiveness.

Based on the aforementioned indicators and their calculated weights (Table 1), a composite health index score was computed for each respondent. This score served as the core dependent variable in the subsequent Regression Disconti-nuity in Time (RDiT) analysis.

**3.1.2. Analysis of measurement results.** The data presented in Table 2 indicate that residents' health outcomes in China have undergone a notable upward trend and a structural transformation. From 2016 to 2022, the average annual growth rate of overall health performance reached 1.2%, with a particularly marked increase between 2018 and 2020. From a compositional perspective, the health status component exhibited steady improvement, with an average annual

**Table 1. Design and weights of health performance evaluation indicator system.**

| Level 1 | Level 2 | weight | Level 3 | direction | weights | Interpretation of indicators |
|---|---|---|---|---|---|---|
| Health Performance | Performance | 0.558 | Self-assessed health | + | 0.177 | represents individual residents' self-rated health based on responses to the survey question: "How healthy do you consider yourself to be?" The scores are coded as 1 = unhealthy, 2 = fair, 3 = fairly healthy, 4 = very healthy, 5 = very healthy. |
| | | | Sudden illness | – | 0.249 | represented by responses to the survey question: "In the past two weeks, have you been unwell?" Reponses are coded as: Yes = 1, No = 0. The severity of the illness is also considered through the question: "How serious did you feel the illness was?" |
| | | | Chronic Illness | – | 0.132 | Represented by responses to the survey question: "In the past six months, have you been diagnosed with a chronic illness diagnosed by a doctor?" Responses are coded as: Yes = 1, No = 0. |
| | Effective | 0.442 | Total Medical Expenditures | – | 0.186 | Calculated as the logarithm of total medical expenditures. |
| | | | Out-of-Pocket Spending | – | 0.187 | Calculated as the logarithm of out-of-pocket medical expenditures. |
| | | | Incidence of Major Medical Expenditures | – | 0.069 | Represents whether a household's annual medical expenditures exceed 40$ of its total annual income. Responses are coded as: Yes = 1, No = 0. |

**Table 2. Mean scores of national population health performance.**

| Indicator | 2016 | 2018 | 2020 | 2022 | Average annual rate of change |
|---|---|---|---|---|---|
| **Health performance** | 0.614 | 0.626 | 0.673 | 0.675 | +1.2% |
| **Performance** | 0.367 | 0.368 | 0.387 | 0.389 | +0.6% |
| **Effective** | 0.247 | 0.258 | 0.286 | 0.286 | +1.8% |

growth rate of 0.6%. Meanwhile, the efficiency component, represented by medical expenditure indicators, demonstrated a more rapid increase, averaging 1.8% annually, which was approximately three times the rate of health status improvement. This trend underscores the strong and immediate impact of health insurance payment reform on medical expenditure efficiency.

As shown in Fig 1, the national and regional health outcome trends from 2016 to 2022 were largely consistent. All regions experienced an initial decline from 2016 to 2018, followed by a marked post-policy increase after the 2019 implementation and a stabilization phase between 2020 and 2022. Regionally, Eastern and Central China consistently outperformed the national average, whereas Western China remained below the national level. Notably, the Western region showed the most significant improvement during the policy period. Despite regional differences in baseline performance, all areas demonstrated measurable enhancements in health performance driven by policy interventions. This improvement was accompanied by a gradual narrowing of regional disparities over time.

Fig 2 tracks the evolution of population health sub-indicators, namely "performance" and "effectiveness". While performance indicators follow similar trajectories across Eastern, Central, and Western China, persistent regional disparities remain throughout the study period with no signs of convergence. In contrast, effectiveness indicators reflecting differences in regional health expenditure levels display significant initial variations. However, after the implementation of health insurance medication payment policies, the gap in effectiveness across regions began to narrow, suggesting a convergence trend over time.

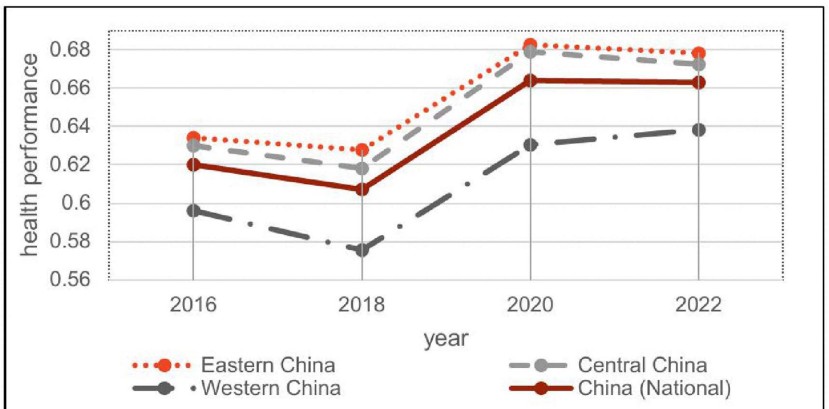

**Fig 1. Trends in population health performance at the national level and across the eastern, central, and western regions, 2016–2022.**

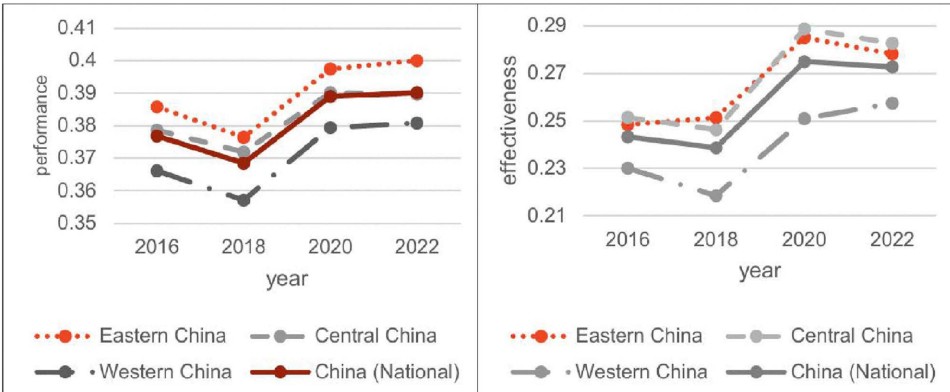

**Fig 2. Trends in population "performance" and "effectiveness" at the national level and across the eastern, central, and western regions, 2016–2022.**

### 3.2. Empirical model setting

**3.2.1. Time breakpoint regression model.** To estimate the causal impact of health insurance medication payment policies on population health outcomes, this study employed the RDiT model originally proposed by Hausman and Rapson [22]. This model was designed to accommodate time-series data by using the timing of policy intervention as the discontinuity point. By segmenting the time axis and observing whether key outcome variables changed abruptly around the policy breakpoint, the model enabled causal inference between the policy and its effects.

Given that the policy was implemented uniformly and significantly across provinces and cities, the chosen breakpoint for this study was the year in which the policy was launched. Specifically, January 2019, when volume-based procurement reform and national drug catalog negotiations were officially rolled out, was used as the breakpoint. Subsequent empirical analysis defined the treatment variables based on this temporal marker.

$$D_i = \begin{cases} 1, T_i \geq 0 \\ 0 \ T_i < 0 \end{cases}$$

where $T_i$ is the driving variable, indicating the monthly interval between the time and the "tipping point" of the policy implementation; $T_i \geq 0$ indicates that the time is after January 2019; $T_i < 0$ indicates that the time is before January 2019; and $D_i$ is the treatment variable, indicating whether provinces and municipalities in China are affected by the policy reform.

The average treatment effect (ATE) of the policy is defined as the difference in the expected value of the sample's dependent variable immediately before and after the breakpoint. The model specification is as follows:

$$\alpha ATEatc = \lim_{\varepsilon \to 0} E\left[Y_{1i}|X_i = c + \varepsilon\right] - \lim_{\varepsilon \to 0} E\left[Y_{oi}|X_i = c + \varepsilon\right] \tag{1}$$

where αATEatc is the average treatment effect; $Y_{oi}$ is the control group individuals before the break date; and $Y_{1i}$ is the individuals treated by the policy after the break date.

The panel model was set up as follows:

$$Y = \alpha_0 + \alpha_1 D + \alpha_2 f(X - t) + \alpha_3 D \times f(X - t) + \alpha_4 controls + \mu \tag{2}$$

where Y is the health performance indicator; D is a treatment variable for the Medicare medication payment management policy mix, which takes the value of 0 before the policy is implemented and the value of 1 after the policy is implemented; x is a driver variable indicating the month in which the surveyed resident is located; f(x-t) is a higher-order polynomial of (x); controls is a control variable; and μ is a random perturbation term.

**3.2.2. bandwidth and polynomial order.** The selection of a 30-month bandwidth for our Regression Discontinuity in Time design was principled, aiming to ensure a robust estimate of the local average treatment effect. This decision directly addresses the limitation of the data-driven MSE-optimal bandwidth (h = 12.8 months), which provides an insufficient sample size and compromises statistical power. Sensitivity analysis confirms that the treatment effect estimate is remarkably stable in both magnitude and significance across a range of bandwidths from 24 to 42 months (see detailed results in Table 11). Positioning our choice at h = 30 within this stability plateau allows us to optimally reduce estimation variance without introducing substantial bias, thus improving the bias-variance trade-off. Furthermore, a 30-month window is theoretically aligned with the policy's implementation timeline, which involves complex adjustments such as hospital formulary updates and changes in prescribing practices. This duration is sufficient to capture the stabilized medium-term policy effect while minimizing contamination from short-term fluctuations or unrelated long-term trends. In summary, the 30-month bandwidth strengthens the statistical power and robustness of our findings while maintaining the local nature of the RDiT estimates.

To determine the appropriate polynomial order, we estimated a series of parametric models for comparison (see detailed results in Table 13). The quadratic model was selected for our baseline nonparametric RDD specification because it provides an optimal balance between model fit and economic interpretability. It yields a positive and significant policy effect that is consistent with theoretical expectations while avoiding the potential overfitting suggested by the counterintuitive result of the cubic model. This choice ensured a robust and theoretically coherent estimation of the treatment effect.

### 3.3. Data sources and variable definitions

**3.3.1. Data sources.** The data used in this study were sourced primarily from CFPS conducted by the China Center for Social Science Surveys at Peking University. Micro-level panel data from four CFPS waves (2016–2022) were used. Due to extensive missing data on children's health indicators, the analysis was restricted to insured individuals aged 16 and above. Data on individual medication expenditures were obtained from the China Health and Wellness Statistical Yearbook. Firm-level variables were extracted from the Wind database, whereas drug import and export data were derived from the General Administration of Customs of China.

## Ethical compliance

This study uses de-identified public data from the CFPS, accessed in January 2025. The analysis complied with all ethical standards for secondary data analysis. The original CFPS project was approved by the Biomedical Ethics Committee of Peking University (IRB No. IRB00001052–14010). Written informed consent was obtained from all original participants: respondents aged ≥16 years signed independently, and for those aged ≤15 years, consent was provided by their guardians. The data were accessed from fully anonymized repositories with no identifiable information. (Consent form template: [CFPS Website] (http://www.isss.pku.edu.cn/cfps/news/index.htm))

**3.3.2. Variable definition.**

**3.3.3. Descriptive statistics.** Table 4 presents the descriptive statistics for the variables in Table 3.

**3.3.4 Multicollinearity Test.** Prior to the formal regression analysis, we calculated the Variance Inflation Factor for all control variables to assess potential multicollinearity issues. As shown in Table 5, the VIF values for all variables were well below the common threshold of 10, and the mean VIF was 4.01. This indicates that severe multicollinearity was not present among the explanatory variables, ensuring the accuracy and stability of the subsequent estimation results.

## 4. Empirical analysis

### 4.1. Breakpoint regression premise hypothesis testing

Breakpoint regression relies on the satisfaction of three key conditions. First, the running variable should be exogenous. This was tested using McCrary's density tests. Following Gao and Pack [23], the running variable in this study was calendar time, which was inherently exogenous within the framework of time-based breakpoint regression. The assignment of time was determined objectively by year and was not subject to the residents' subjective choices. Therefore, manipulation around the policy breakpoint was unlikely to satisfy the first condition. Second, discontinuity should exist in the outcome

**Table 3. Definition of variables.**

| Category Variable Explanation | Variable Name | Variable Explanation |
|---|---|---|
| Explained Variables | Health performance | Comprehensive construction (see Table 3.1 for details) |
| Explanatory Variables | Medical insurance drug payment management policy portfolio | Before the implementation of the policy takes the value of 0, after the implementation of the policy takes the value of 1 |
| Intermediary variables | R&D input | Ln (R&D input amount – million yuan) |
| | R&D output | ln (total number of patent applications by enterprises) |
| | Amount of drug imports | by province, take the logarithm, unit: yuan |
| | Amount of drug exports | by province, take the logarithm, unit: yuan |
| Control Variables | Household | Registration Rural = 0, Urban = 1 |
| | Alcohol consumption | Non-drinking = 0, drinking = 1 |
| | Marital status | No spouse = 0, with a spouse = 1 |
| | per capita disposable income of all residents in the province | RMB |
| | Number of medical and health institutions in the province | Per |
| | Number of practicing physicians in the province | million people |

 

**Table 4. Descriptive statistics.**

|  | Obs | Mean | Std.Dev | Min | Max |
|---|---|---|---|---|---|
| Health Performance<br>Provincial GDP per capita<br>Provincial disposable income | 92097 | 0.634 | 0.258 | 0.007 | 1.000 |
| Months from policy inception | 92097 | 1.852 | 26.524 | −30.000 | 48.000 |
| Household registration | 92097 | 0.509 | 0.500 | 0.000 | 1.000 |
| Alcohol consumption | 92097 | 0.141 | 0.348 | 0.000 | 1.000 |
| Marital status | 92097 | 0.798 | 0.402 | 0.000 | 1.000 |
| Provincial disposable income per capita of all residents | 92097 | 28344.285 | 12264.883 | 13639.000 | 79610.000 |
| Number of health care institutions in the province | 92097 | 45272.774 | 24210.573 | 4254.000 | 90194.000 |
| Number of practicing physicians in the province | 92097 | 12.986 | 6.775 | 0.480 | 28.920 |

**Table 5. Variance Inflation Factor Test Results.**

| Variable | VIF |
|---|---|
| Number of health care institutions in the province | 4.58 |
| Number of practicing physicians in the province | 4.47 |
| Provincial disposable income per capita of all residents | 2.09 |
| Household registration (Hukou) | 1.08 |
| Marital status | 1.01 |
| Alcohol consumption | 1.01 |
| Mean VIF | 4.01 |

variables at the breakpoint. This was tested by fitting regression lines on both sides of the breakpoint. As shown in Fig 3, a significant upward shift in population health performance was observed at the breakpoint, confirming that the second condition was fulfilled.

Third, no discontinuity should occur in the covariates at the breakpoints. Table 6 presents the results of placebo tests using covariates as outcome variables. These results showed no significant jumps, thereby satisfying the third condition.

### 4.2. Regression results

Based on the established indicator system, regression analyses were conducted separately for "health performance", "performance", and "effectiveness". The estimation results presented in Tables 7 and 8 indicate that policy intervention exerted a significant positive impact on population health performance. From a dimensional perspective, the policy demonstrated heterogeneous effects. After controlling for province fixed effects, linear time trends, and relevant covariates, the overall health outcomes of residents increased by 3.4% at the 1% significance level. Specifically, performance improved by 1.3%, while effectiveness increased by 2.5%, suggesting a more pronounced enhancement in expenditure-related efficiency than in health status.

To further evaluate the specific policy impacts, breakpoint regressions were performed on the individual "performance" and "effectiveness" indicators (Table 9). After policy implementation, the health metrics related to "performance" showed statistically significant improvements. Meanwhile, the "effectiveness" indicators demonstrated a substantial reduction in total medical expenditures. These findings suggest that the policy effectively promoted the intensive allocation of medical resources by regulating drug prices and improving reimbursement mechanisms, thereby confirming Hypothesis 1. The empirical results validate the effectiveness of the policy in enhancing health performance and optimizing medical resource

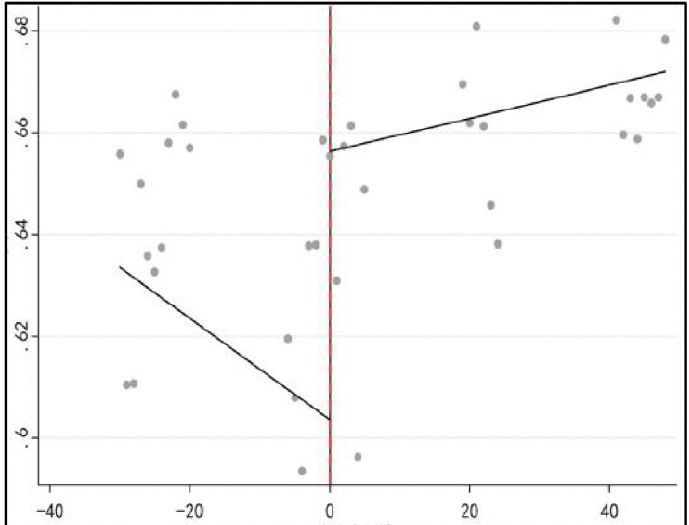
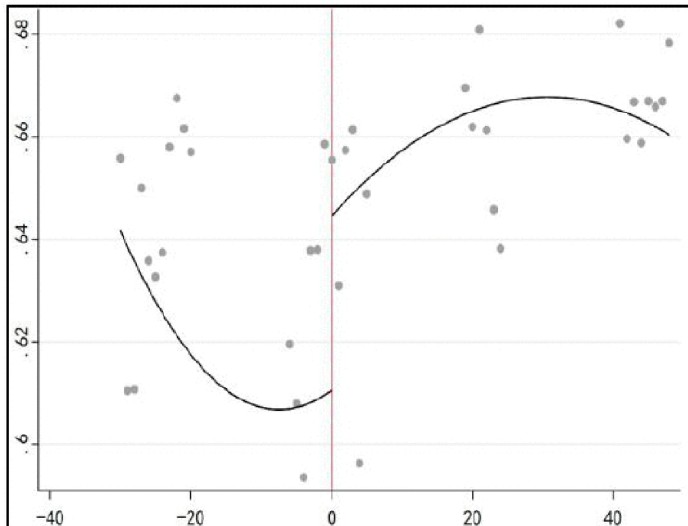

**Fig 3. Changes in population health performance before and after breakpoints (left: primary fit; right: secondary fit).**

**Table 6. Covariate continuity tests for breakpoint regression.**

|  | (1) | (2) | (3) | (4) | (5) | (6) |
|---|---|---|---|---|---|---|
|  | Household registration | Whether Drinking alcohol | Marriage Status | Disposable income | Number of medical and health institutions | Number of practicing physicians |
| RD | −0.027 | −0.013 | −0.021 | −240.654 | 310.767 | 0.374 |
|  | (0.019) | (0.013) | (0.016) | (419.376) | (859.477) | (0.245) |
| Obs | 92097 | 92097 | 92097 | 92097 | 92097 | 92097 |

Notes: 1. Significance levels: * p<0.1, ** p<0.05, *** p<0.01 (These are used consistently across all tables.) 2. Bandwidth=30, polynomial order=2 (These are applied to all subsequent tables.)

**Table 7. Breakpoint regression estimation of population health performance.**

|  | (1) | (2) | (3) |
|---|---|---|---|
|  | Health performance |  |  |
| RD | 0.032*** | 0.041*** | 0.034*** |
|  | (0.009) | (0.010) | (0.010) |
| Control variables | No | No | Yes |
| Province fixed effects | No | Yes | Yes |
| Linear time trend | No | Yes | Yes |
| Obs | 92,097 | 92,097 | 92,097 |

allocation. However, the reduction in out-of-pocket spending was not statistically significant. This result may be explained by the fact that the initial policy focus was on optimizing the allocation of health insurance funds rather than reducing overall spending. The savings generated through centralized procurement were largely redirected to finance high-priced innovative drugs included in the national reimbursement list, with the aim of improving both medication quality and health

**Table 8. Breakpoint regression estimation of "performance" and "effectiveness" on population health.**

|  | (1) | (2) | (3) | (4) | (5) | (6) |
|---|---|---|---|---|---|---|
|  | performance | | | effectiveness | | |
| RD | 0.018*** | 0.015** | 0.013* | 0.014*** | 0.025*** | 0.025*** |
|  | (0.006) | (0.007) | (0.007) | (0.005) | (0.006) | (0.006) |
| Control variables | No | No | Yes | No | No | Yes |
| Province fixed effects | No | Yes | Yes | No | Yes | Yes |
| Linear time trend | No | Yes | Yes | No | Yes | Yes |
| Obs | 92,097 | 92,097 | 92,097 | 92,097 | 92,097 | 92,097 |

**Table 9. Regression results for component indicators of population health performance.**

|  | (1) | (2) | (3) | (4) | (5) | (6) |
|---|---|---|---|---|---|---|
|  | Self-assessment Level of health | Sudden illness | Chronic illness | Total medi-cal expenses | Out-of-pocket spending | Incidence of major medical expenses |
| RD | 0.155*** | −0.003 | −0.026** | −632.664** | −285.339 | −0.034** |
|  | (0.047) | (0.022) | (0.013) | (261.657) | (178.820) | (0.014) |
| Obs | 92,097 | 92,097 | 92,097 | 92,096 | 92,097 | 92,097 |

outcomes. Consequently, policy dividends initially accrued more directly to the supply side of the healthcare system, with a potential time lag before translating into relief of patients' financial burden. Therefore, future policy refinements should emphasize ensuring that cost-containment achievements are more directly reflected in reduced financial pressure for individuals.

### 4.3 Analysis of regional heterogeneity

Table 10 presents the region-specific impacts of the policy on health performance indicators. In Eastern China, although the indicators were positive, they were statistically insignificant, suggesting a marginal effect saturation due to already mature healthcare systems. Central China showed positive but insignificant results for both composite and performance indicators, whereas effectiveness indicators showed a significant improvement, indicating that the policy effectively reduced medical expenditure in this region. In contrast, Western China exhibited significant positive changes across all dimensions. These improvements are attributed to the enhanced accessibility of primary care and increased availability of medications in this resource-constrained region. The findings suggest that the policy combination is the most effective in improving health performance, where baseline medical resources are the weakest. Meanwhile, marginal returns diminish in more developed eastern provinces, and expenditure-oriented benefits are more pronounced in central regions.

## 5. Robustness test

### 5.1. Bandwidth selection sensitivity test

Bandwidth selection is a critical factor that influences the unbiasedness and validity of the breakpoint regression model. By estimating the model across different bandwidth settings, the effect of the intervention can be evaluated over various time horizons. As shown in Table 11, although the estimate was statistically insignificant at the data-driven MSE-optimal bandwidth (h = 12.8), the effective sample size to the right of the cutoff at this bandwidth was relatively small (N = 1,191), leading to high estimation variance and limited statistical power. When employing wider bandwidths that provided more

**Table 10. Separate regression results for population health performance by region.**

|  | (1) | (2) | (3) | (4) | (5) | (6) |
|---|---|---|---|---|---|---|
|  | Eastern Region | | | Central Region | | |
|  | Health Performance | Performance | effectiveness | Health Performance | Performance | effectiveness |
| RD | 0.014 | 0.009 | 0.005 | 0.022 | 0.002 | 0.023** |
|  | (0.023) | (0.014) | (0.012) | (0.016) | (0.009) | (0.010) |
| Obs | 29,661 | 29,661 | 29,661 | 34,707 | 34,707 | 34,707 |
|  | (7) | (8) | (9) | | | |
|  | Western Region | | | | | |
|  | Health Performance | Performance | effectiveness | | | |
| RD | 0.058*** | 0.039*** | 0.019** | | | |
|  | (0.017) | (0.012) | (0.008) | | | |
| Obs | 27,729 | 27,729 | 27,729 | | | |

**Table 11. Regression results for different bandwidths.**

|  | (1) | (2) | (3) | (4) | (5) |
|---|---|---|---|---|---|
|  | h = 12.8 | h = 24 | h = 30 | h = 36 | h = 42 |
| RD Estimate | 0.020 | 0.041*** | 0.032*** | 0.044*** | 0.049*** |
|  | (0.032) | (0.010) | (0.012) | (0.012) | (0.012) |
| Eff. N (Left) | 24,584 | 28,892 | 55,608 | 55,889 | 55,889 |
| Eff. N (Right) | 1,191 | 18,392 | 18,654 | 18,654 | 18,952 |

adequate sample sizes on both sides of the cutoff, the coefficient became highly significant. More importantly, the point estimates exhibited a remarkable degree of stability and a positive trend as the bandwidth increased. This pattern provides strong evidence that our core conclusion is robust to bandwidth selection.

### 5.2. Kernel function selection sensitivity test

The selection of kernel function affects the precision of local average treatment effect (LATE) estimation by altering the assignment of weights to observations near the breakpoint. Table 12 (Models 1 and 2) demonstrates that the main findings remained consistent when different kernel functions such as Epanechnikov and Uniform were used, indicating the robustness of the model specification.

### 5.3. Order choice sensitivity tests

Model (3) in Table 12 shows that the breakpoint estimates remained significantly positive when a first-order polynomial was used. This finding confirms that health performance improved even under different polynomial order assumptions, thereby supporting the robustness of the policy effectiveness hypothesis.

### 5.4. Placebo test

Placebo tests based on hypothetical policy shocks (January 2017 and January 2021) were conducted to ensure the credibility of the 2019 policy effect. The results of local linear regressions across these artificial breakpoints (Table 12, Models 4 and 5) revealed statistically insignificant estimates, confirming that the observed effects were not driven by spurious correlations. These findings validate the causal effect of the actual policy intervention in 2019.

**Table 12. Results of different orders and placebo tests.**

| | (1)<br>Epanechnikov | (2)<br>Uniform | (3)<br>P=1 | (4)<br>year=2017 | (5)<br>year=2021 |
|---|---|---|---|---|---|
| **RD** | 0.043*** | 0.060*** | 0.041*** | 0.001 | −0.008 |
| | (0.009) | (0.009) | (0.008) | (0.017) | (0.017) |
| **Obs** | 92,097 | 92,097 | 92,097 | 92,097 | 92,097 |

## 5.5. Parametric OLS estimates

As an additional robustness check for our nonparametric RDD approach, this section presents the results from parametric OLS models to assess the sensitivity of our findings to functional form specifications. The results (Table 13) indicate that the estimated policy effect was significantly positive under both linear and quadratic trend specifications, which is fully consistent with our baseline nonparametric estimates. Although the cubic model achieved a better fit to the data, its negative coefficient suggested potential overfitting. Accordingly, the quadratic specification yielded a significant, positive, and theoretically coherent estimate of the policy effect while maintaining model parsimony. This result further corroborates our decision in Section 3.2.2 to adopt a quadratic polynomial for the nonparametric RDD estimation. The consistent positive and statistically significant results from both parametric (OLS) and nonparametric (RDD) approaches reinforce the robustness of our main conclusions.

## 5.6. Index reconstruction based on principal component analysis

To examine the sensitivity of our findings to the method of index construction, this study employed Principal Component Analysis (PCA) as an additional robustness check. PCA constructs a new composite index with objective weights by extracting primary common information from the original indicators through dimensionality reduction. The analysis showed that the first three principal components had eigenvalues greater than 0.9 and collectively explained 77.55% of the total variance, effectively capturing the information contained in the original indicators. A composite PCA score was generated using the variance contribution rates of these components as weights. Correlation analysis revealed a very high correlation ($r=0.9406$) between the PCA score and the original index based on the entropy weight method, providing preliminary evidence of consistency between the two measurement methods.

For a direct comparison, both the entropy-weighted index and the PCA-based index were used separately as dependent variables under identical RDiT model specifications. The results reported in Table 14 indicate that the policy effects estimated by both methods were significantly positive at the 1% level. These findings provide strong evidence that the positive impact of the volume-based procurement policy on residents' health outcomes is robust to the choice of index construction method, thereby underscoring the reliability of our main conclusions.

**Table 13. Model selection for polynomial order in parametric RD estimation.**

| Variable | Linear (p=1) | Quadratic (p=2) | Cubic (p=3) |
|---|---|---|---|
| **Policy Effect (D)** | 0.055*** | 0.066*** | −0.020* |
| | (0.005) | (0.007) | (0.011) |
| **Adjusted R-squared** | 0.0229 | 0.0235 | 0.0249 |
| **Akaike (AIC)** | 9883.0 | 9827.7 | 9695.2 |
| **Bayesian (BIC)** | 9977.3 | 9940.8 | 9827.3 |
| **Observations** | 92,097 | 92,097 | 92,097 |

**Table 14. Comparison of different health performance index construction methods.**

|  | (1) | (2) |
|---|---|---|
|  | **PCA** | **entropy weight method** |
| **RD** | 0.164*** | 0.034*** |
|  | (0.038) | (0.010) |
| **Obs** | 92096 | 92097 |

## 6. Discussion of drug prices and mechanisms of action that affect quality

Focusing on volume-based procurement and national drug negotiations, China's medication payment policies utilize pricing mechanisms to reshape physician and patient behaviors. This is reflected in the shift in prescription patterns toward innovative and generic drugs, resulting in reduced medication costs and accelerated pharmaceutical innovations. This study established a unified framework that integrates medication expenditure and drug innovation to evaluate the policy's impact on population health performance. The analysis first investigated how drug price reductions influence health outcomes by generating cost savings for patients. Subsequently, this study explored policy-induced innovation effects by synthesizing R&D data and import-export indicators across domestic and international markets. This dual-pathway analysis revealed the mechanisms by which payment reforms improved both healthcare efficiency and health outcomes.

### 6.1. Expenditures on medicines for the population

Using provincial-level medication cost data (2016–2022) from the China Health Statistical Yearbook, combined with household survey–based health indicators, a breakpoint regression was conducted to estimate the impact of medication payment policies. Fixed-effects models were further applied to assess the influence of outpatient drug expenditure on health outcomes.

As illustrated in Figs 4–7, the shares of outpatient and inpatient drug expenditures in total medical costs consistently declined year-on-year across provinces.

Table 15 confirms that the policy mix significantly reduced per capita outpatient drug costs, demonstrating effective cost containment. However, the results of Model (4) indicated a positive association between inpatient drug expenditures and overall medical costs. This outcome was attributed to the expansion of insurance coverage for high-cost treatments, particularly for cancer and other severe conditions, under the national negotiation framework. This expanded inclusion has led to short-term surges in utilization and temporary increases in costs. Supporting evidence shows that the number of "nationally negotiated drugs" under the dual-channel mechanism grew from 221 in 2020–430 in 2024. For example, in Jiangsu Province, the total sales of such drugs rose from less than 4 billion yuan in 2020 to over 14.6 billion yuan by 2024, benefiting more than 3.6 million patients in 2020 and expanding to over 23 million patients by 2024.

Regarding pharmaceutical cost reductions and their impact on health performance, the total medical expenditure captured under the effectiveness dimension can implicitly encompass drug costs. Therefore, this analysis isolated and examined pharmaceutical expenditures specifically in relation to performance outcomes. As shown in Table 16, outpatient and inpatient expenditures exhibited contrasting effects. The outpatient drug costs both per visit and in aggregate demonstrated significant negative correlations with health performance, confirming the effectiveness of outpatient cost control reforms. In contrast, inpatient care exhibited a different pattern. While per capita hospitalization costs were positively correlated with health outcomes, pharmaceutical expenditures within inpatient services showed statistically significant positive associations with health metrics. This apparent paradox suggested that while the systemic containment of hospitalization costs improved population-level health, targeted investment in pharmaceuticals remained essential for enhancing inpatient therapeutic outcomes. Accordingly, reforms in inpatient pharmaceutical management require a more refined approach. Rather than applying uniform cost-cutting measures, such reforms must aim to curb irrational drug use and

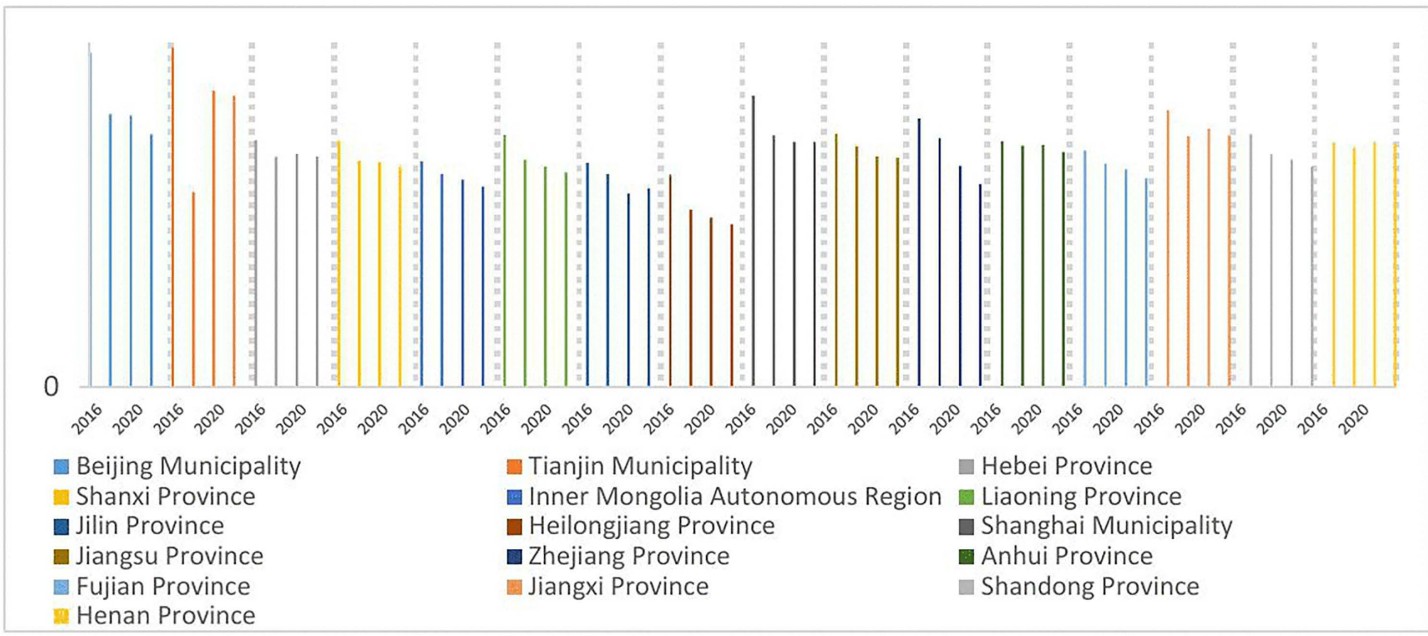

**Fig 4. Average outpatient drug costs per visit as a percentage of total expenditures, 2016–2022.**

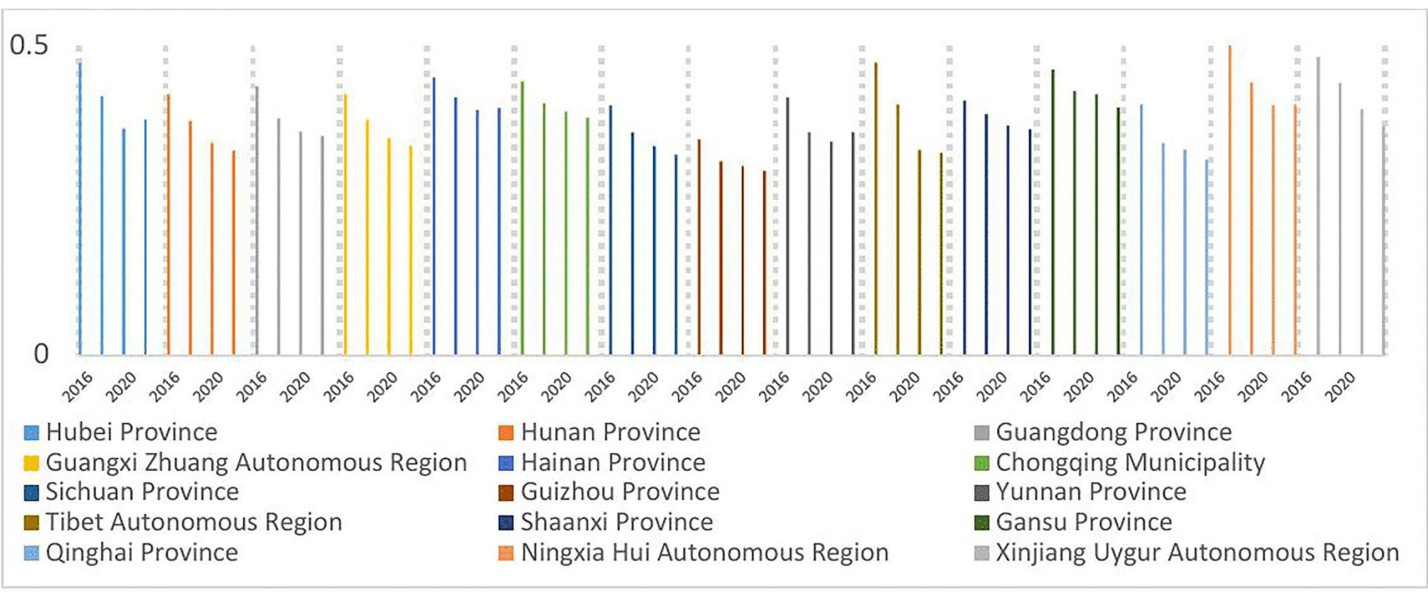

**Fig 5. Average outpatient drug costs per visit as a percentage of total expenditures, 2016–2022.**

inflated pricing while safeguarding access to clinically essential medications. Failure to account for the therapeutic value of specific drugs may compromise patient outcomes, particularly among patients with serious or complex conditions that rely on critical drug therapies.

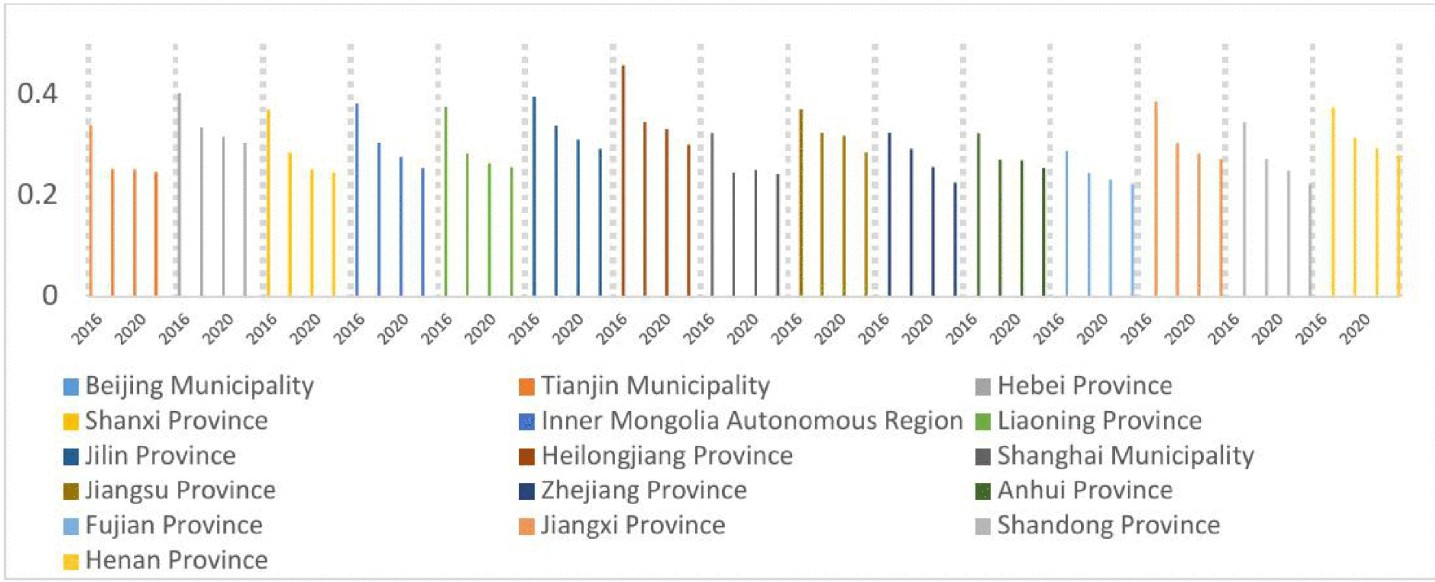

**Fig 6. Average inpatient drug costs per hospitalization as a percentage of total expenditures, 2016–2022.**

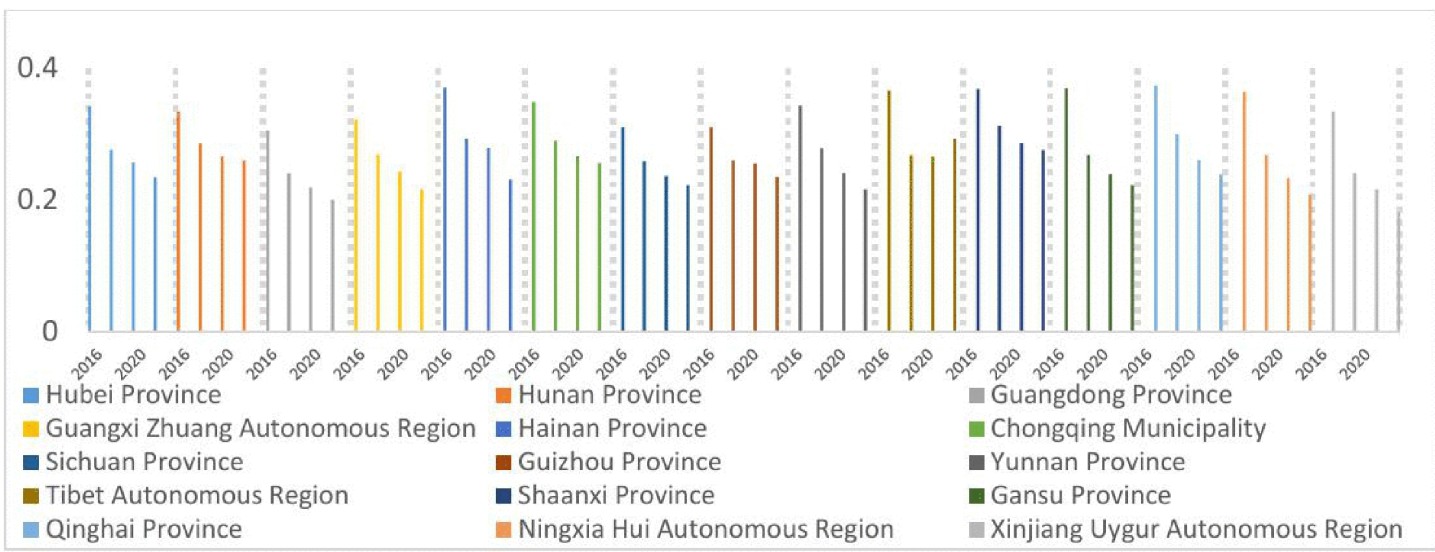

**Fig 7. Average inpatient drug costs per hospitalization as a percentage of total expenditures, 2016–2022.**

## 6.2. Domestic pharmaceutical research and development

Using province-level pharmaceutical innovation data obtained from the Wind database and matched with CFPS data, this study incorporated province-fixed effects to control for regional heterogeneity. R&D input and output indicators were measured using log-transformed R&D expenditures and the number of granted patents, respectively. Breakpoint regression was applied to evaluate the impact of policy implementation on pharmaceutical firms' R&D activities, followed by fixed-effects models to examine how enterprise innovation could affect residents' health performance.

**Table 15. Pharmaceutical cost breakpoint regression results.**

| | (1) | (2) | (3) | (4) |
|---|---|---|---|---|
| | Total average cost per outpatient visit | Average cost of drugs per outpatient visit | Total cost per inpatient visit | Average cost of drugs per inpatient visit |
| RD | −0.031*** | −0.017** | −0.023** | 0.023** |
| | (0.006) | (0.007) | (0.009) | (0.009) |
| Obs | 92097 | 92097 | 92097 | 92097 |

**Table 16. Impact of outpatient pharmaceutical expenditures on population health performance.**

| | (1) | (2) | (3) | (4) |
|---|---|---|---|---|
| | Performance | | | |
| Total average cost per outpatient visit | −0.047*** | | | |
| | (0.009) | | | |
| Average cost of drugs per outpatient visit | | −0.024** | | |
| | | (0.012) | | |
| Total cost per inpatient visit | | | −0.049*** | |
| | | | (0.011) | |
| Average cost of drugs per inpatient visit | | | | 0.034*** |
| | | | | (0.007) |
| Obs | 92097 | 92097 | 92097 | 92097 |

**6.2.1. Firms' R&D inputs.** As shown in Table 17, Model (1) indicates that the implementation of medication payment policy increased provincial pharmaceutical R&D inputs by 5.7%, confirming that the policy effectively strengthened innovation incentives. Models (2) through (4) revealed that each 1% increase in R&D investment led to a 0.027-point improvement in overall health performance, comprising a 0.008-point gain in performance and a 0.019-point gain in effectiveness, both statistically significant at the 1% level. These results suggest that pharmaceutical innovation indirectly enhances health outcomes via technological advancements. While medication cost reductions improve short-term efficiency, sustained improvements in population health require the long-term accumulation of innovation capacity.

**6.2.2. Firms' R&D outputs.** Model (5) in Table 14 indicated that the policy mix increased pharmaceutical R&D output by 7.9%, reflecting a substantial enhancement in innovation productivity. Models (6)–(8) further demonstrated that provincial R&D growth significantly improved population health performance. Specifically, each 1% increase in R&D output led to a 0.011-point rise in composite health scores, with effectiveness gains (0.008) nearly tripling performance

**Table 17. Mechanistic effects of firms' R&D inputs and outputs.**

| | (1) | (2) | (3) | (4) | (5) | (6) | (7) | (8) |
|---|---|---|---|---|---|---|---|---|
| | R&D inputs | Health performance | performance | effectiveness | R&D outputs | Health performance | performance | effectiveness |
| RD | 0.057* | | | | 0.079*** | | | |
| | (0.033) | | | | (0.026) | | | |
| R&D inputs | | 0.027*** | 0.008*** | 0.019*** | | | | |
| | | (0.002) | (0.001) | (0.001) | | | | |
| R&D outputs | | | | | | 0.016*** | 0.004*** | 0.012*** |
| | | | | | | (0.002) | (0.001) | (0.001) |
| Obs | 92002 | 92002 | 92002 | 92002 | 92002 | 92002 | 92002 | 92002 |

 

improvements (0.003). This divergence is consistent with Models (2)–(4), suggesting that pharmaceutical innovation primarily enhances the efficiency of healthcare expenditures rather than directly boosting individual health outcomes.

## 6.3. Import and export of medicines

Using provincial pharmaceutical import–export data from China Customs (2016–2022), which can be matched with household health survey indicators at the province–year level, we first employed a regression discontinuity design to analyze the causal impact of the combined medication payment management policies on pharmaceutical trade, thereby providing direct evidence for the "import substitution–export upgrading" mechanism. Subsequently, a fixed-effects model was applied to examine the pathway through which changes in pharmaceutical imports and exports affect residents' health outcomes.

### 6.3.1. Direct Impact of the Policies on Pharmaceutical Trade.
The results presented in Table 18 Models (1) and Table 19 Models (1) provide direct quantitative evidence of the policy effects: a significant decrease in pharmaceutical import values and a significant increase in pharmaceutical export values following policy implementation. This indicates that volume-based procurement substantially reduced the prices of imported drugs, thereby diminishing market expansion incentives for foreign manufacturers. Concurrently, tender rules favoring domestic generic and innovative drugs significantly curtailed reliance on imported pharmaceuticals, empirically supporting the "import substitution" effect. Moreover, centralized procurement incentivized domestic firms to strengthen their cost control and enhance their international price competitiveness. In parallel, national price negotiations, by providing cash flow support and cost sharing for innovative drugs, enabled companies to accelerate their overseas market expansion, thereby validating the "export upgrading" effect.

This causal interpretation is further illustrated in Fig 8. The fitted lines extrapolated from the pre-policy trend (2016–2018) established a counterfactual trajectory of what would have occurred in the absence of reforms. The post-2019 plots revealed a striking divergence. Actual import values consistently fell below the predicted trend line, visually confirming the

**Table 18. Effects of mechanisms for import of pharmaceuticals.**

|  | (1) | (2) | (3) | (4) |
| --- | --- | --- | --- | --- |
|  | import | Healthperformance | performance | effectiveness |
| RD | −0.354** |  |  |  |
|  | (0.179) |  |  |  |
| import |  | 0.003*** | 0.000 | 0.003*** |
|  |  | (0.001) | (0.000) | (0.000) |
| Obs | 79092 | 79092 | 79092 | 79092 |

**Table 19. Effects of mechanisms for export of pharmaceuticals.**

|  | (1) | (2) | (3) | (4) |
| --- | --- | --- | --- | --- |
|  | export | Healthperformance | performance | effectiveness |
| RD | 0.201** |  |  |  |
|  | (0.092) |  |  |  |
| export |  | 0.022*** | 0.007*** | 0.015*** |
|  |  | (0.002) | (0.001) | (0.001) |
| Obs | 86994 | 86994 | 86994 | 86994 |

In summary, Hypothesis H2 was verified.

significant suppressive effect of the policies on pharmaceutical imports. Conversely, actual export values showed a clear breakout above the forecasted path, providing strong graphical evidence of policy-induced boosts in export competitiveness. This visual demonstration aligns closely with our regression discontinuity findings, thereby reinforcing the validity of both the "import substitution" and "export upgrading" effects of the medication payment reforms.

**6.3.2. Transmission Mechanism from Trade Changes to Health Performance.** Building on the confirmed direct policy effects, we further analyzed the transmission mechanism through which trade changes affect overall health outcomes. Models (2)–(4) of Table 18 showed that each 1% increase in import volume was associated with a marginal improvement of 0.003 units in health performance, driven primarily by gains in effectiveness. This reflects the concentration of imports in high-efficacy therapies, which can enhance resource allocation. However, import dependence remains limited to specific conditions and population groups. Barriers, such as reimbursement constraints and affordability issues, limit broader health gains, suggesting that while import suppression could promote domestic substitution, it may also moderately restrict the scope of population-level health improvements. Models (2)–(4) of Table 19 indicated that each 1% increase in export value contributed to a 0.022-unit increase in overall health performance, with 0.007 attributed to improved health outcomes and 0.015 attributed to enhanced expenditure efficiency. These findings demonstrated that medication payment policies promoted export-driven pharmaceutical innovation, improved drug quality for domestic use, and generated economies of scale that reduced production costs. This exerted indirect downward pressure on domestic healthcare expenditures. In summary, the policy combination supported population health through two channels: improving drug quality and optimizing cost efficiency

## 7. Research conclusion and policy recommendations

### 7.1. Research conclusions

This study constructed a two-dimensional evaluation framework based on "health outcomes" and "medical expenditures" and systematically assessed the impact of medication payment and management policies on population health. The analysis was performed based on breakpoint regression models and leveraged CFPS microdata along with pharmaceutical import-export statistics. The key research findings are summarized as follows:

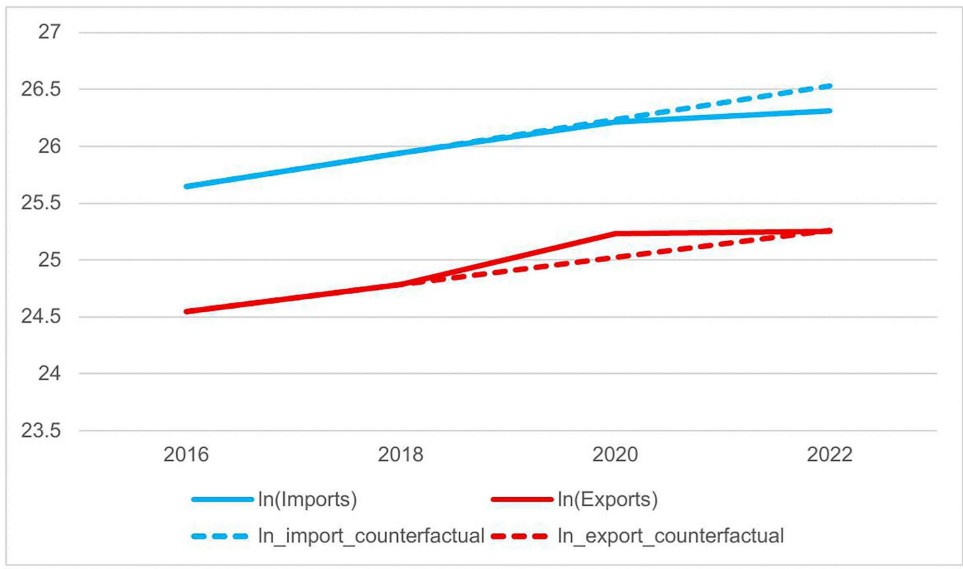

**Fig 8. Trends in China's Pharmaceutical Import and Export Value (2016–2022).**

The policy combination led to a statistically significant 3.4% improvement in the composite health performance index. This overall gain was driven more substantially by a 2.5% enhancement in healthcare expenditure efficiency than by a 1.3% improvement in direct health outcomes. This suggests that within the study period, the policy's initial effect was stronger in optimizing resource use than in directly improving population health status.

The statistically non-significant reduction in out-of-pocket payments underscores the prioritization of resource allocation during the initial phase of reform. Rather than being directly used to lower patient co-payments, substantial savings from volume-based procurement were primarily reinvested into the healthcare system to enhance overall efficiency, for instance, by facilitating access to innovative drugs through national price negotiations. Consequently, policy dividends were initially absorbed by the supply side for systemic optimization, creating a time lag before translating into tangible financial relief for patients. This finding highlights that the short-term priority of the policy was system-level resource reallocation rather than the immediate reduction of patients' financial burdens.

Notably, regional heterogeneity was observed. In Eastern China, the policy effects were statistically insignificant, likely due to saturation in mature healthcare markets. In Central China, the policy primarily improved healthcare expenditure efficiency (i.e., enhancing effectiveness but not performance). Western China achieved significant gains in both health outcomes and expenditure efficiency, underscoring greater policy responsiveness in underdeveloped regions.

Mechanism analysis revealed that the policy enhanced residents' net health gains through three main pathways: reducing out-of-pocket medication costs, incentivizing pharmaceutical R&D investment and innovation, and optimizing medicine trade structures. Improvements were realized by curbing imports and promoting exports, with effectiveness-related gains substantially exceeding performance-related improvements. It is important to emphasize that these findings primarily capture the short- and medium-term effects following policy implementation. Although the observed increase in R&D activity and shifts in trade patterns provide a promising foundation, their long-term sustainability and ultimate impact on industrial transformation require further longitudinal investigation.

### 7.2. Policy recommendations based on empirical evidence

Directly derived from the above findings, we propose the following targeted policy recommendations.

**7.2.1. Regionally differentiated supply strategies to address structural imbalances in medical resources.** Eastern China: Pilot intra-/extra-directory coordinated payment mechanisms should be established to help high-income populations access non-listed innovative drugs through commercial insurance tailored to differentiated needs. Central China: Policy efforts should focus on translating the "cost-reduction effect" of centralized procurement reform into tangible "quality-improvement" outcomes. Key measures include ensuring the stable integration of high-quality pharmaceuticals into the clinical supply system and rigorously standardizing prescribing behaviors to strengthen rational clinical medication. Western China: Fiscal transfers should be increased to expand the scope of collective procurement, ensure a stable supply of essential medicines, and meet the rising demand for rational medication.

**7.2.2. Acceleration of Policy Saving Transmission to Patient Financial Protection.** Future policy efforts should prioritize achieving a natural reduction in patients' out-of-pocket costs through structural reforms, with the central rationale being to position the reform objective as the rationalization, structural optimization, and efficiency enhancement of total medical expenditures. It is essential to maintain the combined strategy of "centralized procurement" and "national price negotiations". By leveraging the synergy of these two instruments, the growth rate of total medical expenditures can be effectively guided toward a rational level or even toward decline. Thus, alleviating patients' financial burden will emerge as an endogenous outcome of enhanced overall system efficiency, rather than as a result imposed by external mandates.

**7.2.3. Establishing a quality-benefit two-dimensional evaluation system and enhancing full-cycle medicine regulation.** For collectively procured drugs, a quality tracing mechanism can be implemented, and the frequency and scope of pharmacological inspections can be enhanced through joint regulatory supervision. Enterprises with repeated inspection failures should be blacklisted. For innovative drugs under insurance coverage, efficacy monitoring should be

conducted every 3–5 years using hospital EMRs and insurance settlement data to assess health outcomes. Based on these evaluations, renegotiation or withdrawal of underperforming products should be initiated.

### 7.3. Prospective implications for long-term development

The short-term empirical findings of this study, particularly the significant increase in pharmaceutical R&D investment and output, provide valuable insights into long-term policy planning. The initial success of the policy mix in stimulating innovation suggests a promising trajectory for fostering an innovation-driven pharmaceutical industry. To sustain this momentum, future policies should consider establishing a more integrated innovation chain, encompassing R&D risk-sharing funds, streamlined market access for breakthrough therapies, and mechanisms to ensure hospital adoption of cost-effective innovations. It is important to emphasize that these are forward-looking implications inferred from early signals. Their realization will require sustained policy commitment and warrants further long-term investigation.

### 7.4. Limitations and future research

This study had several limitations. First, the RDiT design, although robust for identifying causal effects, relies on the strong assumption that no other confounding policies or events occurred around the 2019 breakpoint. Although we controlled for observable factors, unmeasured confounders may have biased the estimates. Second, our data captured only short- to medium-term effects; the long-term sustainability of the observed improvements and their ultimate impact on industrial transformation remain unanswered questions. Future research should draw on longer time series to track these dynamics and employ alternative methods to validate our findings.

### Acknowledgments

The authors would like to express their sincere gratitude to all individuals and organizations that contributed to this research. We are especially grateful to our colleagues at the School of Economics, Wuhan Textile University for their stimulating discussions and constructive feedback. Furthermore, we extend our sincere appreciation to the anonymous reviewers for their meticulous review and thoughtful suggestions, which have significantly improved the quality of this manuscript. Finally, we acknowledge all those who contributed indirectly to this research but whose names are not listed here.

### Author contributions

**Writing – original draft:** Dingqiang Duan, yun yang.

**Writing – review & editing:** Dingqiang Duan.

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
