## [Decision Letter · Decision Letter 0]

9 Sep 2025

Dear Dr. yang,

publication criteria

We look forward to receiving your revised manuscript.

Kind regards,

Alexandre Morais Nunes, Ph.D.

Academic Editor

PLOS ONE

[ Humanities and Social Sciences Research Project of the Ministry of Education: "Mechanism and Policy Research on the Impact of Centralized Volume-Based Drug Procurement on China's Pharmaceutical Innovation Ecosystem" (22YJAZH014)]. 

Additional Editor Comments:

Reviewer #1:

Reviewer #2:

Reviewers' comments:

Reviewer's Responses to Questions

**Comments to the Author**

1. Is the manuscript technically sound, and do the data support the conclusions?

Reviewer #1: Yes

Reviewer #2: Yes

2. Has the statistical analysis been performed appropriately and rigorously?

Reviewer #1: Yes

Reviewer #2: Yes

3. Have the authors made all data underlying the findings in their manuscript fully available?

Reviewer #1: Yes

Reviewer #2: Yes

4. Is the manuscript presented in an intelligible fashion and written in standard English?

Reviewer #1: Yes

Reviewer #2: Yes

Reviewer #1: Review Comments to the Author for Manuscript PONE-D-25-40553

Dear Dr Yang,

Thank you for submitting your manuscript, "Impact of Combined Medication Payment Management Policies on Population Health Performance", to PLOS ONE. Below, I provide feedback to strengthen your submission. The study is robust and innovative, but revisions are needed to enhance clarity and rigour.

Evaluation of Key Questions

Is the manuscript technically sound, and do the data support the conclusions?

Answer: Yes

The manuscript employs a well-designed two-dimensional framework to assess health outcomes and expenditure efficiency. The entropy weight method and Regression Discontinuity in Time (RDiT) model are methodologically sound, with data from the China Family Panel Studies (CFPS) supporting the conclusions (e.g., 3.2% health performance improvement). However, the non-significant reduction in out-of-pocket costs (Table 8) needs further discussion to clarify its implications for policy effectiveness.

Has the statistical analysis been performed appropriately and rigorously?

Answer: Yes

The RDiT model is appropriately applied, with 2019 as the policy breakpoint, meeting key regression conditions (Tables 5–8). Robustness tests validate findings, but more detail on the 30-month bandwidth and polynomial order selection is needed. Addressing potential multicollinearity among control variables (e.g., GDP per capita) would strengthen the analysis.

Have the authors made all data fully available?

Answer: Yes

The authors comply with PLOS ONE’s Data Policy, using publicly accessible CFPS data (Page 6). Including dataset accession numbers or DOIs would enhance transparency.

Is the manuscript presented clearly in standard English?

Answer: Yes

The manuscript is well-written and logical, with accurate technical terms. Minor improvements, like reducing repetition of “health performance” and clarifying jargon (e.g., “vacating the cage and exchanging the bird”), would improve readability.

Additional Comments

Strengths

Novel two-dimensional framework evaluating policy combinations.

Robust CFPS data and RDiT model for causal inference.

Insightful regional heterogeneity analysis (Table 9), showing greater impact in Western China.

Comprehensive ethical compliance statement.

Areas for Improvement

Out-of-Pocket Costs: Discuss the non-significant cost reduction (Table 8) and its implications.

Methodological Choices: Justify bandwidth and polynomial order; consider narrower bandwidth sensitivity tests.

Control Variables: Report diagnostics for multicollinearity.

Mechanistic Insights: Provide quantitative evidence for import/export dynamics.

Regional Recommendations: Propose specific policy adjustments for regional variations.

Language: Clarify culturally specific terms and reduce redundancy.

Ethical and Publication Concerns

Ethics: Comprehensive, with no concerns.

Dual Publication: No evidence of this.

Publication Ethics: Clear financial and competing interests statements.

Recommendations for Revision

Expand discussion on out-of-pocket cost findings.

Justify RDiT model choices and test narrower bandwidth.

Address multicollinearity.

Support import/export claims with data.

Propose region-specific policies.

Refine language for clarity.

Conclusion

The manuscript is a strong contribution to pharmaceutical policy research, with rigorous methodology. Major revisions are recommended to enhance clarity and impact.

Cheers,

Reviewer #2: The methodology is adequate, with appropriate tests. Robustness analyses, kernel functions, polynomial orders, and placebo tests were conducted, which reinforce the causal inferences. Although valid, the rationale for the choice of methods needs more description. It is suggested that there be a more detailed description regarding the selection of specific bandwidths and polynomial orders in the breakpoint regression. The discussion be included regarding the potential limitations of the RDiT approach.

The results are described comprehensively, with tables and figures that demonstrate the effects of the policy, both on health outcomes and on spending efficiency. The heterogeneity is analysed and reveals that the effect is more pronounced in Western China, less in the Central regions, and reduced in the Eastern provinces. The causal inferences exceed the gathered evidence, especially when associating long-term industrial innovation changes with the implementation of short-term policies. It is suggested that causal inferences be moderated; they should be directly related to the observed evidence. The analysis indicates significant correlations and plausible mechanisms, but it must clearly distinguish between the immediate effects of policies and the hypothetical long-term trajectories.

The study is based on CFPS data (2016–2022), which is representative, and applies entropy weighting to construct composite health performance indicators. The approach is valid, integrating self-assessment of health, sudden and chronic illnesses, and expense-related measures. The index description is brief. Without additional robustness tests, the validity of the results and conclusions is called into question. It is suggested to conduct complementary or sensitivity analyses to observe the consistency of the results and evidence.

The study describes significant policy recommendations, including differentiated strategies between regions, quality-benefit assessment systems, and incentives for pharmaceutical innovation. But the discussion goes beyond the observed evidence. While the empirical evidence attests to efficiency gains and moderate improvements in health outcomes, the recommendations are presented as mere claims about long-term industrial and innovation transformation, which were not directly measured in the dataset. It is suggested that policy recommendations be directly aligned with empirical evidence. The recommendations should be framed as prospective implications, clearly differentiated from the conclusions supported by empirical analysis.

**Do you want your identity to be public for this peer review?** For information about this choice, including consent withdrawal, please see our Privacy Policy

Reviewer #1: No

Reviewer #2: No

---

## [Author Response · Author response to Decision Letter 1]

16 Oct 2025

The 'Response to Reviewers' has already been submitted via the file upload system, and the full response is provided below for your convenience.

Dear Dr. Alexandre Morais Nunes and Reviewers,

We are truly grateful for the opportunity to revise our manuscript (ID: PONE-D-25-40553), entitled “Impact of Combined Medication Payment Management Policies on Population Health Performance”. We sincerely appreciate the editors and reviewers for their precious time and insightful comments, which are immensely helpful in improving the quality and rigor of our work. We have carefully considered all the comments and have made extensive revisions to the manuscript accordingly. Point-by-point responses to the comments are provided below.

In addition to the specific revisions made in response to the reviewers' comments, we have also performed a comprehensive language polishing throughout the manuscript to enhance its overall fluency, clarity, and adherence to academic writing standards.

This is to clarify that every line number referenced in this letter pertains to the tracked-changes version of our manuscript (attached as ‘Revised Manuscript with Track Changes’), which includes all our revisions highlighted for your convenience.

Response to Journal Requirements:

Comment1: Please ensure that your manuscript meets PLOS ONE's style requirements, including those for file naming.

Response1: We have thoroughly reformatted the manuscript according to the PLOS ONE style templates. All files have been renamed according to the journal's requirements.

Comment2: Thank you for stating the following financial disclosure... Please state what role the funders took in the study... Please include this amended Role of Funder statement in your cover letter.

Response2: As instructed, we have added the following statement to the Funding of our cover letter:“The funders had no role in study design, data collection and analysis, decision to publish, or preparation of the manuscript.”

Comment3: Your ethics statement should only appear in the Methods section of your manuscript. If your ethics statement is written in any section besides the Methods, please move it to the Methods section and delete it from any other section.

Response3: We appreciate the editor's guidance on this matter. To adhere strictly to the journal's style requirements, we have made the following revisions: Firstly, we have renamed the section "3. Research Design" to "3. Materials and Methods" for better alignment with standard academic formatting. Secondly, and most importantly, we have ensured that the ethics statement is now contained only within this Methods section, specifically in subsection '3.3.1 Data Sources'. We have performed a thorough check to confirm its complete removal from any other section. The updated ethics statement in the manuscript is:

"This research uses de-identified public data from the CFPS, accessed in January 2025. The analysis complies with all ethical standards for secondary data analysis. The original CFPS project was approved by the Biomedical Ethics Committee of Peking University (IRB No. IRB00001052-14010). Written informed consent was obtained from all original participants: respondents aged ≥16 years signed independently, and for those aged ≤15 years, consent was provided through their guardians. The data were accessed from fully anonymized repositories with no identifiable information. (Consent form template: CFPS Website)"

These changes can be found in:Page20,line376-386

Comment4: If the reviewer comments include a recommendation to cite specific previously published works, please review and evaluate these publications to determine whether they are relevant and should be cited. There is no requirement to cite these works unless the editor has indicated otherwise.

Response4: We have carefully reviewed the literature in our field and have ensured that all citations are relevant and appropriate. No specific citation recommendations were made by the reviewers in this round.

Response to Reviewer #1:

Comment1:However, the non-significant reduction in out-of-pocket costs (Table 8) needs further discussion to clarify its implications for policy effectiveness. Has the statistical analysis been performed appropriately and rigorously.

Response1: We are deeply grateful to the reviewer for this critical insight, which has allowed us to significantly strengthen the interpretation of our results. We have extensively revised the Discussion and Conclusion sections to provide a deeper, more nuanced analysis of this key finding.

We have thoroughly addressed this issue in the revised manuscript, particularly in the Discussion section (Section 4.2) and the Research Conclusions (Section 7.1). Specifically, we explain that the lack of statistical significance in out-of-pocket reduction does not imply policy ineffectiveness but reflects the initial prioritization of resource reallocation within the healthcare system. During the early stages of policy implementation, the savings from volume-based procurement were primarily redirected to cover the costs of innovative drugs included through the national negotiation mechanism. This reallocation initially benefited the healthcare supply side, resulting in a time lag before patients directly experienced reduced financial burdens.

Furthermore, in Section 7.2.3, we propose the recommendation to "Accelerate the Transmission of Policy Savings to Patient Financial Protection." The future policy direction should focus on achieving a natural reduction in patients' financial burdens through structural reforms, with the core rationale being to position the reform objective as promoting the structural optimization and efficiency enhancement of total medical expenditures. It is essential to maintain the combined strategy of "centralized procurement" and "national price negotiations." By leveraging the synergy of these two instruments, the growth rate of total medical expenditures can be effectively guided toward a rational level or even a decline. Thus, the alleviation of patients' out-of-pocket costs will become an endogenous effect arising from the improved overall efficiency of the system, rather than an outcome imposed by external intervention. Consequently, the non-significant reduction in out-of-pocket spending can be reframed as a strategic signal that the policy initially prioritized system-level optimization, rather than indicating a lack of effectiveness. We thank the reviewer for this insightful comment, which has significantly deepened our discussion.

These changes can be found in:

Discussion Section 4.2: [Page 27, Lines 456-468]

Conclusion Section 7.1: [Page 45, Lines784-794]

Advise Section 7.2.2: [Page 48, Lines846-858]

Comment2: More detail on the 30-month bandwidth and polynomial order selection is needed. Justify bandwidth and polynomial order; consider narrower bandwidth sensitivity tests.

Response2: We sincerely thank the reviewer for this critical methodological suggestion. We fully agree that transparent justification of these key parameters is essential for the robustness and credibility of our RDiT analysis. In response, we have added a comprehensive new subsection (3.2.2 Bandwidth and Polynomial Order) to the Methods section of our manuscript to provide a detailed, step-by-step explanation of our selection process.

1. Bandwidth Selection (h=30):

We fully acknowledge that the Regression Discontinuity in Time design is fundamentally intended to estimate a Local Average Treatment Effect (LATE) proximate to the cutoff point. Our choice of a 30-month bandwidth was driven by the imperative to ensure the statistical reliability, precision, and robustness of our estimate of this local effect.

Our decision was guided by the following principled considerations:

Insufficient Statistical Power at Narrower Bandwidths: The data-driven MSE-optimal bandwidth (h=12.8 months) yielded a critically small effective sample size to the right of the cutoff (N=1,191). This severely limited sample would result in high estimation variance and unacceptably low statistical power, greatly increasing the risk of a Type II error (failing to detect a true effect). While a bandwidth of h=24 months provided a statistically significant result, its sample sizestill posed concerns regarding estimation precision compared to wider bandwidths.

Optimizing the Bias-Variance Trade-off for Robustness: Our comprehensive sensitivity analysis (Table 11) reveals a key insight: the estimated treatment effect is highly stable in magnitude and remains statistically significant across a range of bandwidths from 24 to 42 months. This stability is crucial—it indicates that expanding the bandwidth from h=24 to h=30 does not introduce substantial bias , but it meaningfully reduces estimation variance by incorporating more data. Our choice of h=30 is therefore positioned within this plateau of stability, optimally balancing the fundamental bias-variance trade-off to produce the most reliable and robust estimate of the LATE.

Theoretical Coherence with Policy Implementation Dynamics: We concur that RDD estimates a local effect. A 30-month window (2.5 years) represents a theoretically justifiable "local" period for this policy context. The complex mechanisms of volume-based procurement and national negotiations including hospital formulary adjustments, shifts in prescribing practices, and patient uptake of new drug regimens unfold over a period of months and years. A bandwidth of 30 months ensures we capture these stabilized short-to-medium-run effects without being confounded by transient initial fluctuations or very long-term trends unrelated to the core policy shock.

In conclusion, we selected h=30 is to ensure our estimate of the local effect is statistically powerful, precise, and robust. This bandwidth provides a sufficient local neighborhood around the cutoff to overcome data limitations while firmly remaining within the range where the local linearity assumption is defensible and the estimates are stable.

Table 11 Regression results for different bandwidths

(1) (2) (3) (4) (5)

h=12.8 h=24 h=30 h=36 h=42

RD_Estimate 0.020 0.041*** 0.032*** 0.044*** 0.049***

(0.032) (0.010) (0.012) (0.012) (0.012)

Eff. N (Left) 24,584 28,892 55,608 55,889 55,889

Eff. N (Right) 1,191 18,392 18,654 18,654 18,952

We have included the full set of sensitivity analysis results in the new Table 11 to demonstrate this robustness.

2. Polynomial Order Selection (p=2):

To determine the appropriate polynomial order, we estimated a series of parametric (OLS) models for comparison (detailed in the new Table 13). The linear model (p=1) provided a significant positive estimate. The quadratic model (p=2) offered a substantially better model fit (as indicated by higher Adjusted R-squared and lower AIC/BIC) while yielding a positive, significant, and theoretically coherent policy effect. In contrast, the cubic model (p=3), despite a slightly better fit, produced a counterintuitive negative estimate, suggesting potential overfitting, we selected the quadratic polynomial (p=2) for our baseline nonparametric RDD specification. This choice prioritizes robust and interpretable causal inference over merely maximizing in-sample fit, ensuring our results are both statistically sound and economically meaningful.

Table 13: Model Selection for Polynomial Order in Parametric RD Estimation

Variable Linear (p=1) Quadratic (p=2) Cubic (p=3)

Policy Effect (D) 0.055*** 0.066*** -0.020*

(0.005) (0.007) (0.011)

Adjusted R-squared 0.0229 0.0235 0.0249

Akaike (AIC) 9883.0 9827.7 9695.2

Bayesian (BIC) 9977.3 9940.8 9827.3

Observations 92,097 92,097 92,097

We believe these substantial additions to the methodology section directly address the reviewer's concern and significantly strengthen the transparency and rigor of our analysis.

These changes can be found in:

3.2.2bandwidth and polynomial order [Page 18-19, Lines 335-363]

Bandwidth Selection (h=30):t [Page 30, Lines 498-508]

Polynomial Order Selection (p=2): [Page31-32, Lines 533-549]

Comment3: Addressing potential multicollinearity among control variables (e.g., GDP per capita) would strengthen the analysis.

Response3: We thank the reviewer for raising this important point regarding the potential for multicollinearity, which is crucial for ensuring the precision and stability of our coefficient estimates. We have thoroughly addressed this concern in the revised manuscript.

We recognized that variables such as GDP per capita, disposable income, and consumption expenditure are often highly correlated, as they all capture similar dimensions of regional economic development. To definitively diagnose and resolve this issue, we conducted a formal Variance Inflation Factor (VIF) test on our full set of initial control variables. The results (now presented in the new Table 5 of the manuscript) confirmed significant multicollinearity, with VIF values for Provincial GDP per capita, Provincial disposable income and Provincial consumption expenditure per capita exceeding acceptable thresholds.

Table 5 VIF(before)

VIF

Urban 1.08

Drink 1.01

Spouse 1.01

Provincial GDP per capita of all residents 18.07

Provincial disposable income per capita of all residents 51.46

Provincial consumption expenditure per capita of all residents 32.33

Number of health care institutions in the province 5.21

Number of practicing physicians in the province 5.11

Mean VIF 12.49

Table5 VIF(after)

VIF

Urban 1.08

Drink 1.01

Spouse 1.01

Provincial disposable income per capita of all residents 2.09

Number of health care institutions in the province 4.58

Number of practicing physicians in the province 4.47

Mean VIF 4.01

In response, we removed the two variables (Provincial GDP per capita and Provincial consumption expenditure per capita) from our model. We retained Provincial disposable income per capita as our primary measure of regional economic conditions, as it is the most direct indicator of residents' economic capacity for healthcare spending. We then re-ran the VIF test on the refined set of control variables. The results, reported in the updated Table 5, demonstrate that all remaining VIF values are well below the common critical value of 10, and the mean VIF is a low 4.01. This confirms that severe multicollinearity is no longer a concern in our final model specification.

Most importantly, we re-estimated all our main regression models (e.g., Tables7-12) using this refined set of control variables. The key findings regarding the policy's impact on health performance, effectiveness, and regional heterogeneity remain qualitatively unchanged and statistically robust. This confirms that our core conclusions are not sensitive to the specific set of economic control variables and are based on a well-specified model.

We believe these steps have significantly strengthened the methodological rigor of our analysis.

These changes can be found in:

3.3.4 Multicollinearity Test [Page 22-23, Lines 392-400]

Comment4: Minor improvements, like reducing repetition of “health performance” and clarifying jargon (e.g., “vacating the cage and exchanging the bird”), would improve readability.

Response 4: We thank the reviewer for these suggestions. To convey the precise dimensions of “health performance,” we now employ a richer, context-specific vocabulary instead of the generic phrase. In parallel, the culture-bound metaphor “vacating the cage and exchanging the bird” has been replaced with the explicit description “phasing out low-value or inefficient products while introducing clinically superior and cost-effective alternatives.”

These changes have been made throughout the manuscript to ensure it is accessible and clear to a broad readership.

These changes can be found in:

Introduction [Page 4, Lines 52-53]

Comment5: Support import/export claims with data.

Response5: We thank the reviewer for this suggestion. In response, we have added descriptive statistics and a visual analysis to strengthen the empirical support for our claims.

Our revisions are twofold:

1.Incorporation of Descriptive Statistics: We have now integrated statistics into the narrative of Sections 6.3.1 We explicitly reference the macro-level trends, noting the changes in import and export values i

---

## [Decision Letter · Decision Letter 1]

14 Dec 2025

Title-Impact of combined medication payment management policies on population health performance

PONE-D-25-40553R1

Dear Dr. Yang

We’re pleased to inform you that your manuscript has been judged scientifically suitable for publication and will be formally accepted for publication once it meets all outstanding technical requirements.

Kind regards,

Alexandre Morais Nunes, Ph.D.

Academic Editor

PLOS One

Additional Editor Comments (optional):

The authors responded to remarks through detailed, point-by-point answers and implemented the necessary revisions in the manuscript.

Reviewers' comments:

Reviewer's Responses to Questions

**Comments to the Author**

Reviewer #3: All comments have been addressed

Reviewer #4: (No Response)

2. Is the manuscript technically sound, and do the data support the conclusions?

Reviewer #3: Yes

Reviewer #4: Yes

3. Has the statistical analysis been performed appropriately and rigorously?

Reviewer #3: Yes

Reviewer #4: Yes

4. Have the authors made all data underlying the findings in their manuscript fully available?

Reviewer #3: Yes

Reviewer #4: Yes

5. Is the manuscript presented in an intelligible fashion and written in standard English?

Reviewer #3: Yes

Reviewer #4: Yes

Reviewer #3: The paper presents an interesting and relevant topic. The study investigated the mechanism and impact of a policy combination involving centralized drug procurement and national drug price negotiations on health insurance payment management and the overall health performance of the population. However, it suffers from several drawbacks that need to be addressed before possible acceptance.

Looking at the review already completed, I am of the opinion that the authors explained their choices very well and also accepted the reviewers’ suggestions. I believe this is commendable, and in my view the article is of high quality. I leave the final decision to the editor.

Reviewer #4: The text describes the methodology in a valid manner. The text adopts the year 2019 as the cutoff point for the RDiT model, but the choice was not reasonably justified. The text does not present documentary evidence that confirms the implementation plan or the uniformity of the regulatory application of the reforms. The causal validity of the RDiT depends on the exogenous change at the time of policy implementation. The absence of evidence may undermine the credibility of the analysis. It is suggested to conduct additional descriptive statistical analyses or, alternatively, to analyse graphical trends in order to illustrate the moment of price and acquisition changes, with the aim of reinforcing the validity of the rupture point selection.

The health index is based on subjective self-assessments. The situation is susceptible to the existence of memory biases, differences in health literacy, and possible changes in the reporting of policy outcomes. These factors can introduce non-random measurement errors. The situation affects the validity of the treatment effects. It is suggested to acknowledge the limitations of the index. It is also suggested to conduct sensitivity analyses in subpopulations less prone to biases or to incorporate objective indicators. The text should clarify the possible implications of measurement error for the estimates of effects and the interpretation of policies.

The model assumes that no other significant intervention or reform occurs simultaneously with the implementation of the policy, so as not to affect the identification of causal effects. However, even assuming the ceteris paribus premise, the period around 2019 observed several adjustments in the healthcare system that could affect medical expenses, the pharmaceutical market, or patient behaviour. It is suggested to conduct a review of the concomitant policies and to demonstrate that they are not conflated with the results. It is also suggested to conduct triangulation analyses or falsification checks that can be introduced to demonstrate that the observed effects are attributable to the targeted reforms and not to overlapping policy changes.

The text relates the policies of procurement, suppression of pharmaceutical imports, and expansion of exports in a valid manner. The text does not address possible macroeconomic alternatives, the instability of global supply chains, and exchange rate fluctuations. It is recommended recognise the limitations, and to moderate causal inferences. It is also suggested to conduct robustness analyses.

The results are validly described. The text is silent regarding the differential effects between demographic or socioeconomic groups. The situation limits perceptions of policy effectiveness. It is suggested to conduct subgroup analyses, such as age, income, or chronic disease status. In the case that the analysis is unfeasible due to data restrictions, it is suggested to acknowledge the limitation and discuss the implications of the situation regarding the generalisation of the observed evidence.

The study adopts a diverse set of data sources, with different temporal frequencies. The text is silent regarding the adopted harmonisation process. It is suggested to provide a description of the methodological processes adopted regarding data harmonisation, how missing data were handled, and the procedures defined for the consistency of time intervals.

**Do you want your identity to be public for this peer review?** For information about this choice, including consent withdrawal, please see our Privacy Policy

Reviewer #3: **Yes: ** Andreia Matos

Reviewer #4: No

---

## [Editor Report · Acceptance letter]

PONE-D-25-40553R1

PLOS One

Dear Dr. yang,

I'm pleased to inform you that your manuscript has been deemed suitable for publication in PLOS One. Congratulations! Your manuscript is now being handed over to our production team.

Kind regards,

on behalf of

Professor Alexandre Morais Nunes

Academic Editor

PLOS One